# Community factors and hospital wide readmission rates: Does context matter?

Erica S. Spatz[1,2]*, Susannah M. Bernheim[2,3], Leora I. Horwitz[4,5,6], Jeph Herrin[1]

**1** Section of Cardiovascular Medicine, Yale School of Medicine, New Haven, CT, United States of America, **2** Yale/Yale New Haven Health Center for Outcomes Research and Evaluation, New Haven, CT, United States of America, **3** Division of Medicine, Yale School of Medicine, New Haven, CT, United States of America, **4** Division of Healthcare Delivery Science, Department of Population Health, NYU School of Medicine, New York, NY, United States of America, **5** Center for Healthcare Innovation and Delivery Science, NYU Grossman School of Medicine New York, NY, United States of America, **6** Division of General Internal Medicine and Clinical Innovation, Department of Medicine, NYU School of Medicine, New York, NY, United States of America

* Erica.spatz@yale.edu

## Abstract

### Background

The environment in which a patient lives influences their health outcomes. However, the degree to which community factors are associated with readmissions is uncertain.

### Objective

To estimate the influence of community factors on the Centers for Medicare & Medicaid Services risk-standardized hospital-wide readmission measure (HWR)–a quality performance measure in the U.S.

### Research design

We assessed 71 community variables in 6 domains related to health outcomes: clinical care; health behaviors; social and economic factors; the physical environment; demographics; and social capital.

### Subjects

Medicare fee-for-service patients eligible for the HWR measure between July 2014-June 2015 (n = 6,790,723). Patients were linked to community variables using their 5-digit zip code of residence.

### Methods

We used a random forest algorithm to rank variables for their importance in predicting HWR scores. Variables were entered into 6 domain-specific multivariable regression models in order of decreasing importance. Variables with P-values <0.10 were retained for a final model, after eliminating any that were collinear.

**Data Availability Statement:** Data is available at (https://doi.org/10.3886/E122901V1) Ann Arbor, MI: Inter-university Consortium for Political and Social Research [distributor], 2020-09-28.

**Funding:** This study was funded by a grant from the Agency for Healthcare Research and Quality, titled: Understanding Hospital Readmission Rates: Patient, Hospital and Community Effects, awarded to SMB, LIH, and JH (1 R01 HS022882 01).

**Competing interests:** The authors have declared that no competing interests exist.

## Results

Among 71 community variables, 19 were retained in the 6 domain models and in the final model. Domains which explained the most to least variance in HWR were: physical environment ($R^2$ = 15%); clinical care ($R^2$ = 12%); demographics ($R^2$ = 11%); social and economic environment ($R^2$ = 7%); health behaviors ($R^2$ = 9%); and social capital ($R^2$ = 8%). In the final model, the 19 variables explained more than a quarter of the variance in readmission rates ($R^2$ = 27%).

## Conclusions

Readmissions for a wide range of clinical conditions are influenced by factors relating to the communities in which patients reside. These findings can be used to target efforts to keep patients out of the hospital.

## Introduction

Readmission following a hospital discharge represents a potentially avoidable outcome and has been a key focus of recent policy, research and quality improvement efforts [1–4] Accordingly, the Centers for Medicare & Medicaid Services (CMS) publicly report a global measure of readmission following an index hospitalization for virtually any condition (hospital-wide readmission [HWR]) [4] This measure is risk-adjusted for patient-level clinical factors which influence the risk of readmission. However, social risk factors may also contribute to readmission, both on an individual level, such as individuals' socioeconomic position, race, ethnicity and culture, gender orientation and sexuality; as well as on a community level, including factors in a patients' home environment. While the relationship between individual-level social risk factors and outcomes has garnered much attention [5–9], less is known about the relative contribution of community-level social, economic and physical factors to outcomes. Communities are multi-dimensional and have attributes that may interact in multiple ways to influence the risk of patients being readmitted soon after discharge [10–20]. Understanding the relative importance of neighborhood characteristics on individual health outcomes will help policymakers identify and prioritize community-based interventions to improve outcomes that might provide benefits beyond purely hospital-focused activities.

Several community factors may influence a patient's risk for readmission. For example, some community factors like accessibility of primary care providers, availability of transportation, healthy foods and places to exercise, density of non-profit organizations, and accessible, high-quality acute rehabilitation centers and skilled nursing facilities may directly support or hinder recovery and thus may contribute to patients' risk of readmission [16, 18, 21]. Additionally, the socioeconomic and social environment, defined by factors such as poverty, unemployment and crime, may exert an indirect effect, as these social ills may result in fewer resources and competing priorities for health [10, 13, 15, 17, 22]. Assessments of the contribution of these and other community factors to hospital readmissions is important for informing health system and community-based efforts to lower readmission rates.

Accordingly, we adapted and expanded the Robert Wood Johnson Foundation (RWJF) County Health Rankings framework to examine how community factors may relate to hospital readmission rates [23]. The original framework consists of 4 domains: clinical care; health behaviors; social and economic factors; and physical environment. We expanded this

framework to also include the domains of demographics and social capital. We viewed these 6 domains as representing different 'spheres' of influence with different effect sizes and implications for causality and intervention. Specifically, clinical care features of a community can influence readmissions through both quality and access to care; the RWJF County Health Rankings framework captures measures of both. Community healthy behaviors, measured at the population level, capture residents' healthy activities as well as the availability of open spaces, fresh fruits and vegetables–reflecting the dominant culture towards health and available health resources that may make it more or less easy for a patient returning home from the hospital to engage in health-promoting activities. The social and economic domain includes metrics of education, income, and employment which may directly or indirectly influence patient risk wherein a person has less available resources or the community is deprived of resources which are necessary for recovery and health (e.g., available funds to pay for transportation to appointments; caregiver who can take time off of work; community support groups). The physical environment–including variables such as housing density and air and water quality– is known to affect individual health—impacting respiratory and cardiovascular physiology and increasing people's risk for admission. To these four domains from the RWJF County Health Rankings framework we added two additional community domains which we hypothesized would influence the risk of readmission. One was a demographic domain, which includes age and race/ethnicity characteristics of the community; we hypothesized that communities with greater percentages of minority residents have more limited access to care and poorer quality of care, beyond what is already captured in the HWR measure, because of historical disparities [24]. and that other socioeconomic factors (e.g., employment; income; education) have different effects in minority populations [25–27]. The other additional domain was social capital, which includes metrics on voter turnout and the number of active non-profits and religious organizations [28]; we hypothesized that communities with greater social cohesion and citizen involvement would also provide additional support to patients who might otherwise return to the hospital.

We had 2 primary objectives: to identify variables most associated with hospital readmission rates and to estimate the relative contribution of distinct spheres or domains of influence as well as specific community variables within each domain hypothesized to impact hospital readmission rates. Importantly, and distinct from prior work, we focus on patients admitted for a broad range of different clinical conditions; additionally, we measure the attributes of the communities where patients reside, not where the hospital is located [18], which may be more relevant to patient outcomes [29]. Ultimately, a better understanding of the community factors that have an impact on post-discharge outcomes is essential to promote the development of effective, safe and patient-centered healthcare environments.

## Methods

### Design and setting

The context for this study was the United States. Medicare is a federal program, providing insurance coverage to people aged 65 years and older and who are on long-term disability. The Centers for Medicare and Medicaid Services, through which the Medicare program is administered, also generates hospital quality performance metrics which are publicly reported, and which are tied to value-based care programs. We assessed U.S. based community factors, as described below, in relation to a hospital quality measure of readmissions.

Specifically, we assessed the contribution of community-level variables to performance on the measure of HWR, a measure of 30-day readmissions among Medicare fee-for-service patients hospitalized for all but a few conditions or procedures. This measure was previously

developed by members of our group and is currently publicly-reported by CMS [4]. We identified a wide range of community variables which we linked to zip code of patient residence, and applied a machine learning approach, random forest, to identify which variables had the most 'importance' (a parameter assigned by random forest to each variable) for predicting hospital readmission rates. Since variables from different domains may correlate with each other (e.g., poverty and smoking), which may impact statistical significance though not necessarily clinical significance, we first organized all top ranked variables into their respective domains and assessed the relative importance of variables within each domain; the most important variables from each domain were then retained for final comprehensive assessment.

**Participants.** Patients included in the hospital measure of all-cause readmissions are Medicare fee-for-service beneficiaries aged 65 years and older who are hospitalized for a condition in one of the following specialty cohorts: medicine, surgery/gynecology, cardiorespiratory, cardiovascular, and neurology. Among these patients, we assessed the hospital risk-standardized readmission rate, as described below. We also assessed patients' community context, based on the patients' 5-digit zip code; also described in more detail below.

**Independent study variables: Community factors.** The variables for each domain were selected based on the RWJF County Health Rankings, prior literature and/or face validity, and were drawn from several sources, including the U.S. Census Bureau, Bureau of Health Professions Area Health Resources File, Northeast Regional Center for Rural Development Social Capital Index, Nursing Home Compare and Home Health Compare [23, 30]. Each variable was assessed at the smallest unit for which there were data, ranging from Census block to county. See Table 1 for a list of the 71 community variables assessed for this analysis, and the level of measurement.

**Dependent study variable: Hospital wide readmission measure.** The HWR measure, previously described [4], was used as the main outcome for this analysis, since this measure captures the broadest range of patients who are hospitalized and at-risk for readmission. Briefly, the HWR measure is a volume-weighted logarithmic mean of the standardized readmission rates (SRRs) of 5 specialty cohorts: surgical, medical, cardiovascular, cardiorespiratory and neurological. The readmission risk ratio of each cohort is calculated as the ratio of the predicted readmissions divided by the expected readmissions. The numerator is the sum of the predicted probability of readmission for all readmissions, estimated using a hierarchical logistic regression model that includes a hospital-specific effect estimated for each hospital as well as covariates for age, principal diagnosis and comorbidity. The denominator is the same as the numerator but replaces the hospital-specific effect in the model with the average hospital-specific effect of all hospitals in the sample. Only readmissions that are deemed as unplanned are included in the outcome. To exclude planned readmissions, the HWR uses a "planned readmission" algorithm which identifies procedures that are: 1) "always" or "potentially" planned procedures; and 2) not associated with an acute medical discharge diagnosis code; we used version 4.0 of this algorithm [31]. For analyses we used the logarithmic mean SRR. We used 2015–2016 Medicare Part A to identify the cohort and to assess the outcome for the measurement period July 1, 2015 through June 30, 2016.

**Analysis.** We linked all variables to Medicare FFS patients included in the HWR measure using their 5-digit zip code of residence. For zip codes that crossed more than one county, if only county-level data were available we used the value for the county with the greatest population. For each hospital we then averaged the factor value over all patients discharged from that hospital and categorized these means into quintiles based on their distribution across hospitals. We then summarized hospital HWR SRR for each factor quintile, reporting mean and standard deviation.

Our analytic approach focused on identifying key variables both within each domain and across all domains. Therefore, we used a multi-step analysis. First, because of the large number

**Table 1. Variables included in study, by domain.**

| Domain | Variable | Source | Level | P-value | R2 |
|---|---|---|---|---|---|
| Demographic | | | | | |
| | 2011 population estimate Value | RWJF County Health Rankings | FIPS | <0.001 | 0.0355 |
| | % below 18 years of age | RWJF County Health Rankings | FIPS | <0.001 | 0.0159 |
| | % aged 65 years and older | RWJF County Health Rankings | FIPS | <0.001 | 0.0050 |
| | % non-Hispanic African American | RWJF County Health Rankings | FIPS | <0.001 | 0.0801 |
| | % Am Indian or AK Native | RWJF County Health Rankings | FIPS | <0.001 | 0.0320 |
| | % Asian | RWJF County Health Rankings | FIPS | <0.001 | 0.0197 |
| | % Hawaiian/Pacific Islander | RWJF County Health Rankings | FIPS | <0.001 | 0.0301 |
| | % Hispanic | RWJF County Health Rankings | FIPS | 0.005 | 0.0029 |
| | % non-Hispanic White | RWJF County Health Rankings | FIPS | <0.001 | 0.0234 |
| | % non-English Value | RWJF County Health Rankings | FIPS | <0.001 | 0.0122 |
| Health Behaviors | | | | | |
| | Adult smoking Value | RWJF County Health Rankings | FIPS | <0.001 | 0.0060 |
| | Adult obesity Value | RWJF County Health Rankings | FIPS | <0.001 | 0.0151 |
| | Food environment index Value | RWJF County Health Rankings | FIPS | <0.001 | 0.0097 |
| | Physical inactivity Value | RWJF County Health Rankings | FIPS | <0.001 | 0.0298 |
| | Access to exercise opportunities Value | RWJF County Health Rankings | FIPS | <0.001 | 0.0511 |
| | Excessive drinking Value | RWJF County Health Rankings | FIPS | 0.060 | 0.0014 |
| | Alcohol-impaired driving deaths Value | RWJF County Health Rankings | FIPS | <0.001 | 0.0344 |
| | Sexually transmitted infections Value | RWJF County Health Rankings | FIPS | <0.001 | 0.0117 |
| | Teen births Value | RWJF County Health Rankings | FIPS | 0.001 | 0.0037 |
| | Food insecurity Value | RWJF County Health Rankings | FIPS | <0.001 | 0.0214 |
| | Limited access to healthy foods Value | RWJF County Health Rankings | FIPS | <0.001 | 0.0181 |
| | Motor vehicle crash deaths Value | RWJF County Health Rankings | FIPS | <0.001 | 0.0228 |
| | Drug poisoning deaths Value | RWJF County Health Rankings | FIPS | 0.003 | 0.0033 |
| Clinical Care | | | | | |
| | Uninsured Value | RWJF County Health Rankings | FIPS | <0.001 | 0.0060 |
| | Primary care physicians Value | RWJF County Health Rankings | FIPS | 0.026 | 0.0019 |
| | Dentists Value | RWJF County Health Rankings | FIPS | 0.017 | 0.0021 |
| | Mental health providers Value | RWJF County Health Rankings | FIPS | 0.954 | -0.0009 |
| | Diabetic screening Value | RWJF County Health Rankings | FIPS | <0.001 | 0.0078 |
| | Mammography screening Value | RWJF County Health Rankings | FIPS | <0.001 | 0.0075 |
| | Nursing Home Quality: Vaccines | Nursing Home Compare | ZIP | <0.001 | 0.0058 |
| | Nursing Home Quality: Pain | Nursing Home Compare | ZIP | <0.001 | 0.0645 |
| | HHA Quality: Improvement | Home Health Compare | ZIP | <0.001 | 0.0160 |
| | HHA Quality: Pain, bedsores | Home Health Compare | ZIP | <0.001 | 0.0072 |
| | HHA Quality: Vaccines | Home Health Compare | ZIP | <0.001 | 0.0826 |
| | # Nursing Home beds | Area Health Resources File | FIPS | <0.001 | 0.0444 |
| | # Hospital Beds | Area Health Resources File | FIPS | <0.001 | 0.0388 |
| | GPs/100k | Area Health Resources File | FIPS | <0.001 | 0.0669 |
| | Med Specialists/100k | Area Health Resources File | FIPS | <0.001 | 0.0410 |
| | Surg Specialists/100k | Area Health Resources File | FIPS | <0.001 | 0.0156 |
| | Other Specialists/100k | Area Health Resources File | FIPS | <0.001 | 0.0153 |
| | GPs/Specialists | Area Health Resources File | FIPS | <0.001 | 0.0535 |
| | # Fed Qual Health Centers | Area Health Resources File | FIPS | <0.001 | 0.0242 |
| Social & Economic Environment | | | | | |
| | High school graduation Value | RWJF County Health Rankings | FIPS | <0.001 | 0.0238 |

*(Continued)*

**Table 1.** (Continued)

| Domain | Variable | Source | Level | P-value | R2 |
|---|---|---|---|---|---|
| | Some college Value | RWJF County Health Rankings | FIPS | 0.020 | 0.0020 |
| | Unemployment Value | RWJF County Health Rankings | FIPS | <0.001 | 0.0507 |
| | Children in poverty Value | RWJF County Health Rankings | FIPS | <0.001 | 0.0237 |
| | Income inequality Value | RWJF County Health Rankings | FIPS | <0.001 | 0.0679 |
| | Children of single parents Value | RWJF County Health Rankings | FIPS | <0.001 | 0.0397 |
| | Violent crime Value | RWJF County Health Rankings | FIPS | <0.001 | 0.0392 |
| | Injury deaths Value | RWJF County Health Rankings | FIPS | <0.001 | 0.0195 |
| | % Commute car | American Community Survey | FIPS | <0.001 | 0.0345 |
| | % Commute bike | American Community Survey | FIPS | <0.001 | 0.0144 |
| | % Commute work at home | American Community Survey | FIPS | <0.001 | 0.0265 |
| | % Commute public | American Community Survey | FIPS | <0.001 | 0.0316 |
| | % Preschool | American Community Survey | FIPS | <0.001 | 0.0186 |
| | % Kindergarten | American Community Survey | FIPS | <0.001 | 0.0342 |
| | Median Home Value | American Community Survey | FIPS | <0.001 | 0.0060 |
| Physical Environment | | | | | |
| | Particulate air population Value | RWJF County Health Rankings | FIPS | <0.001 | 0.0224 |
| | Drinking water violations Value | RWJF County Health Rankings | FIPS | <0.001 | 0.0087 |
| | Severe housing problems Value | RWJF County Health Rankings | FIPS | <0.001 | 0.0449 |
| | Driving alone to work Value | RWJF County Health Rankings | FIPS | <0.001 | 0.0181 |
| | Long commute—driving alone Value | RWJF County Health Rankings | FIPS | <0.001 | 0.0711 |
| | Housing Density | Area Health Resources File | FIPS | <0.001 | 0.0532 |
| | Good Air Days | Area Health Resources File | FIPS | 0.001 | 0.0038 |
| | Daily Fine Particulate Matter | Area Health Resources File | FIPS | <0.001 | 0.0222 |
| | Toxic Sites | Area Health Resources File | FIPS | <0.001 | 0.0099 |
| | Housing Units | American Community Survey | FIPS | <0.001 | 0.0413 |
| Social Capital | | | | | |
| | Social Capital: Associations | Social Capital Index | FIPS | <0.001 | 0.0388 |
| | Social Capital: Voter turnout | Social Capital Index | FIPS | <0.001 | 0.0277 |
| | Social Capital: Census response | Social Capital Index | FIPS | <0.001 | 0.0257 |
| | Social Capital: Non-profits | Social Capital Index | FIPS | <0.001 | 0.0312 |

*All variables were stratified into quintiles.

**P-values and $R^2$ are from a bivariate regression model, weighted for volume.

of candidate community variables and potential for interaction effects, we used a random forest (RF) algorithm to identify the variables with greatest influence on risk standardized readmissions. The RF approach is robust to interactions, missing data, and collinearity, using an ensemble of small classification trees to construct a 'forest' which best predicts the outcome [32]. It is also self-validating in that each tree is trained on a new random half sample and validated on the second half. Our RF incorporated a linear regression model with hospital standardized risk ratio as the outcome to classify variables; we constructed 10,000 trees, each using 3 randomly selected variables. The final RF was used to assign an importance score to each variable, representing the proportion of decision paths which ran through that variable.

Though the random forest algorithm can rank variables on importance, it cannot provide estimates of independent effects for each variable. Therefore, in the second step, for each domain we identified the most important factor from the RF results and estimated a domain specific model using that variable. We then sequentially added the next most important

variables in that domain, assessing the change in adjusted $R^2$, and retaining additional variables that increased the $R^2$ by a (relative) 10% or more. The rationale for estimating separate domain models is to identify key variables within each domain which may have less relative importance when assessing effects across all domains. Because variables from different domains might be highly correlated, we assessed the variance decomposition matrix for all retained variables, in order to eliminate any variables that were redundant (as indicated by a condition number of at least 20 and a common variance portion greater than 0.50) [33] These retained variables were entered into a final multivariable model, which collectively assessed the key variables from each domain for an overall assessment of the relative contribution of community variables to readmission rates. All regression models were hospital level models, weighted for the readmission denominator volume, and for each we report the proportion of variance explained $R^2$ and the overall Wald P-value for each variable.

The Institutional Review Boards of Yale School of Medicine and New York University School of Medicine approved the study. All analysis was performed using SAS 9.3 (SAS Institute Inc, Cary NC, USA), Stata 15.1 (StataCorp, College Station TX, USA), R 3.5.2 (R Foundation for Statistical Computing, Vienna, Austria), and the R package *randomForestSRC*.

**Ethical considerations.** This study was conducted with de-identified data of the patients and the hospitals.

## Results

After excluding patients with a planned index admission, who were transferred to another hospital, or who died, enrolled in hospice or left against medical advice during the index admission, the final cohort was 6,790,723 discharges from 4,711 hospitals. The mean (SD) risk standardized readmission rate was 15.3% (0.81%) and the median (interquartile range) was 15.3% (14.8%– 15.8%). Among the 71 community variables, the RF algorithm and sequential model selection identified 19 variables for inclusion in the domain specific models. The 10 most important factors associated with higher readmissions in the random forest model were (in descending order of importance): increasing percent of population that is non-Hispanic African American; lower rate of vaccination by home health agencies; poorer nursing home quality related to pain management; severe housing problems; greater physical inactivity; lower density of social associations; greater income inequality; children living in poverty; lower Census response rates; and greater housing density (Fig 1).

The domains that explained the most to least variation were: physical environment ($R^2$ for the domain specific model = 15%); clinical care ($R^2$ = 12%); demographic ($R^2$ = 11%); social and economic environment ($R^2$ = 7%); health behaviors ($R^2$ = 9%); and social capital ($R^2$ = 8%). Of the 19 variables, none were eliminated by the variance decomposition assessment. Thus all 19 variables were included in the final model (Table 2); all variables except voting participation were significant; together these contextual factors explained more than a quarter of the variance in hospital HWR readmission rate ($R^2$ = 27%).

## Discussion

In this comprehensive examination of a range of contextual factors and their association with all-cause clinically risk-adjusted hospital readmission rates we found that a range of community variables across 6 distinct domains were independently associated with readmission rates; the 19 community variables which were most associated with readmission rates explained 27% of the variation in hospital-wide readmissions. Importantly, among the variables most accounting for this variation, many could be modifiable by local, state and federal governments, community organizations, hospitals, and health care systems. For example, these findings

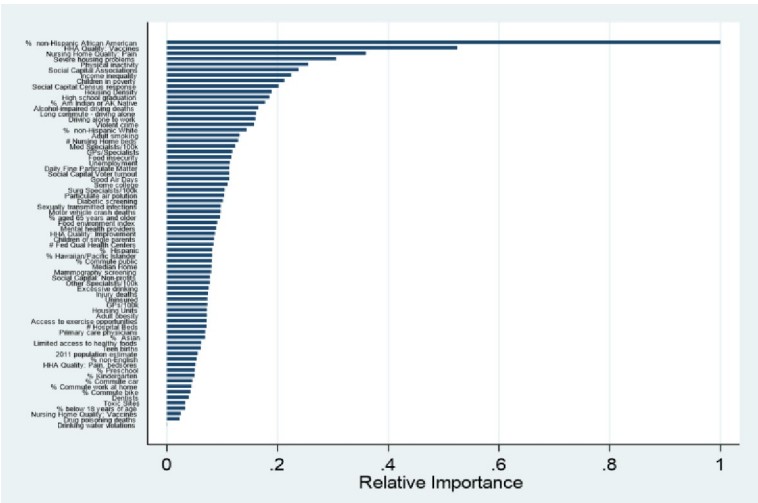

**Fig 1. Results of random forest analysis, showing relative importance of each factor for predicting hospital overall Standardized Readmission Ratio (SRR).**

could give impetus for healthcare delivery systems to partner with community agencies to improve the quality of home care, or to partner with local community health centers to improve vaccination rates. Next, as previously demonstrated, demographics and social and economic factors independently contributed to outcomes, including the proportion of the community that was African American or Native American or Alaskan, as well as the unemployment rate, the level of income inequality and the percent of children living in poverty. These associations amplify known racial and economic disparities–each of which needs to be separately considered. While race is associated with lower socioeconomic status, it is also the case that racial minorities derive less benefits from higher socioeconomic status; i.e., racial disparities in health outcomes persist even among high income and highly educated groups [25, 34, 35]. This is known as the Minorities' Diminished Return (MDR) theory, which can serve as a compass for addressing racial and ethnic disparities in health outcomes [27].

We also found that housing density and commuting, and people's social connectedness and engagement were significant. This finding provides evidence to support investments in social services by health payors. Payors, such as Medicaid, are increasingly allowing waivers that encourage patients to use premiums towards social issues; hospitals could lobby for such waivers [36]. Moreover, hospitals are becoming more involved in the composition and quality of the medical community in which their patients live [21, 37, 38]. For example, some hospitals are partnering with nursing homes with a track record of high-quality performance and quality improvement initiatives [39]. Whether these efforts will result in reduced readmission rates should be a focus of future study.

Finally, we found that 3 variables indicating communities' social capital (Census response rates, voter turnout and density of social associations) explained 8% of the variation in readmission rates. Unfortunately, we do not have detailed data available about the types of social supports existing in these communities. However, in extrapolating these findings we might infer that in communities where residents have limited family or friend support, and limited engagement with religious institutions and community centers, there may be a need for hospitals or other health entities to furnish outpatient supports. Patients discharged from the hospital are weak and convalescing–and providing assistance with basic meals and shopping, transportation to appointments, along with emotional support needs to be considered.

**Table 2. Final multivariable model.**

| Final Model* | | | | |
|---|---|---|---|---|
| **Variable** | **Value*** | **Mean (SD)** | **P-value** | **R2** |
| | | | | 0.2658 |
| Housing Density | | | 0.012 | |
| | Q1 | ref | | |
| | Q 2 | -0.202 (0.100) | | |
| | Q 3 | -0.266 (0.104) | | |
| | Q 4 | -0.359 (0.111) | | |
| | Q 5 | -0.296 (0.120) | | |
| Daily Fine Particulate Matter | | | <0.001 | |
| | Q 1 | ref | | |
| | Q 2 | 0.192 (0.056) | | |
| | Q 3 | 0.159 (0.060) | | |
| | Q 4 | 0.018 (0.064) | | |
| | Q 5 | -0.126 (0.066) | | |
| # Nursing Home beds | | | <0.001 | |
| | Q 1 | ref | | |
| | Q 2 | 0.076 (0.060) | | |
| | Q 3 | 0.177 (0.060) | | |
| | Q 4 | 0.074 (0.062) | | |
| | Q 5 | 0.270 (0.064) | | |
| HHA Quality: Vaccines | | | <0.001 | |
| | Q 1 | ref | | |
| | Q 2 | -0.153 (0.047) | | |
| | Q 3 | -0.195 (0.050) | | |
| | Q 4 | -0.250 (0.056) | | |
| | Q 5 | -0.333 (0.062) | | |
| Nursing Home Quality: Pain | | | 0.004 | |
| | Q 1 | ref | | |
| | Q 2 | 0.053 (0.049) | | |
| | Q 3 | -0.097 (0.055) | | |
| | Q 4 | -0.002 (0.060) | | |
| | Q 5 | 0.057 (0.069) | | |
| % non-Hispanic White | | | <0.001 | |
| | Q 1 | ref | | |
| | Q 2 | 0.011 (0.052) | | |
| | Q 3 | 0.182 (0.064) | | |
| | Q 4 | 0.293 (0.079) | | |
| | Q 5 | 0.588 (0.104) | | |
| Alcohol-impaired driving deaths | | | <0.001 | |
| | Q 1 | ref | | |
| | Q 2 | 0.107 (0.049) | | |
| | Q 3 | 0.067 (0.050) | | |
| | Q 4 | 0.022 (0.051) | | |
| | Q 5 | -0.123 (0.057) | | |
| Severe housing problems | | | 0.015 | |
| | Q 1 | ref | | |
| | Q 2 | 0.090 (0.069) | | |

(*Continued*)

**Table 2.** (Continued)

| Final Model* | | | | |
|---|---|---|---|---|
| **Variable** | **Value*** | **Mean (SD)** | **P-value** | **R2** |
| | Q 3 | 0.108 (0.077) | | |
| | Q 4 | 0.238 (0.087) | | |
| | Q 5 | 0.300 (0.102) | | |
| Long commute—driving alone | | | <0.001 | |
| | Q 1 | ref | | |
| | Q 2 | 0.142 (0.062) | | |
| | Q 3 | 0.250 (0.065) | | |
| | Q 4 | 0.344 (0.068) | | |
| | Q 5 | 0.298 (0.076) | | |
| Food insecurity | | | <0.001 | |
| | Q 1 | ref | | |
| | Q 2 | -0.175 (0.051) | | |
| | Q 3 | -0.311 (0.053) | | |
| | Q 4 | -0.273 (0.059) | | |
| | Q 5 | -0.067 (0.072) | | |
| Income inequality | | | <0.001 | |
| | Q 1 | ref | | |
| | Q 2 | 0.070 (0.054) | | |
| | Q 3 | 0.085 (0.057) | | |
| | Q 4 | 0.265 (0.061) | | |
| | Q 5 | 0.232 (0.071) | | |
| % non-Hispanic African American | | | <0.001 | |
| | Q 1 | ref | | |
| | Q 2 | 0.159 (0.077) | | |
| | Q 3 | 0.347 (0.086) | | |
| | Q 4 | 0.587 (0.095) | | |
| | Q 5 | 0.537 (0.108) | | |
| % American Indian or Alaskan Native | | | 0.014 | |
| | Q 1 | ref | | |
| | Q 2 | -0.022 (0.048) | | |
| | Q 3 | -0.149 (0.051) | | |
| | Q 4 | -0.135 (0.063) | | |
| | Q 5 | -0.113 (0.076) | | |
| Driving alone to work | | | 0.004 | |
| | Q 1 | ref | | |
| | Q 2 | -0.132 (0.055) | | |
| | Q 3 | -0.075 (0.062) | | |
| | Q 4 | -0.038 (0.067) | | |
| | Q 5 | 0.074 (0.075) | | |
| Physical inactivity | | | <0.001 | |
| | Q 1 | ref | | |
| | Q 2 | 0.219 (0.053) | | |
| | Q 3 | 0.396 (0.062) | | |
| | Q 4 | 0.343 (0.076) | | |
| | Q 5 | 0.434 (0.093) | | |
| Adult smoking | | | 0.004 | |

(*Continued*)

**Table 2.** (Continued)

| Variable | Value* | Mean (SD) | P-value | R2 |
|---|---|---|---|---|
| **Final Model*** | | | | |
| | Q 1 | ref | | |
| | Q 2 | -0.050 (0.051) | | |
| | Q 3 | 0.080 (0.061) | | |
| | Q 4 | 0.079 (0.072) | | |
| | Q 5 | 0.211 (0.084) | | |
| Social Capital: Associations | | | <0.001 | |
| | Q 1 | ref | | |
| | Q 2 | -0.278 (0.049) | | |
| | Q 3 | -0.460 (0.062) | | |
| | Q 4 | -0.403 (0.070) | | |
| | Q 5 | -0.361 (0.088) | | |
| Social Capital: Voter turnout | | | 0.354 | |
| | Q 1 | ref | | |
| | Q 2 | 0.047 (0.051) | | |
| | Q 3 | 0.007 (0.054) | | |
| | Q 4 | 0.013 (0.055) | | |
| | Q 5 | -0.061 (0.062) | | |
| Social Capital: Census response | | | 0.046 | |
| | Q 1 | ref | | |
| | Q 2 | -0.101 (0.064) | | |
| | Q 3 | 0.018 (0.067) | | |
| | Q 4 | -0.074 (0.073) | | |
| | Q 5 | -0.085 (0.084) | | |

*Values represent quintiles for each variable.

The observation that social risk factors contribute to readmission rates is not new, but prior studies have not assessed the range of factors explored in this study and the differential effects of their impact on readmission. In a national study of hospital readmission rates, Herrin et al assessed 3 community contextual domains: socioeconomic status; access to care; and nursing home status [18]. They found that 58% of the variation in performance was explained by these factors, with the most potent factor being access to care. Other studies have examined singular domains. For example, a study by Pandolfi et al also found that a community's nursing home quality is associated with hospital readmission rates [20] and a study by Brewster et al found that social capital was associated with readmission rates [10]. Our study attempts to offer a fuller picture of how a community's well-being can impact health outcomes.

As part of its implementation of the IMPACT Act of 2014, the U.S. Department of Health and Human Services is charged with providing analyses and recommendations on considering social risk within CMS payment programs. Moreover, as of 2019, CMS is applying differential penalties in reimbursement based on poor hospital performance on quality measures, according to the proportion of low SES patients at hospitals. Global measures of community-level SES are being considered in value-based care programs, including the Accountable Care Organization program and the Merit-based Incentive Payment System program. Indeed many accountable care organizations are already working to address community factors such as housing quality and transportation, with the goal of reducing health care costs and improving quality outcomes [40, 41]. Understanding the effects of a comprehensive set of contextual

factors across a range of healthcare environments and settings is important for coordinating efforts to address these factors and reducing readmissions [42].

These findings have several limitations. Community level variables may not represent the reality of an individual's living situation or community. We selected the most granular level of variable collection, though we still expect there to be heterogeneity within each unit. Still, these variables are intended to inform a hospital's readmission rates, and not to predict the readmission rate for any one individual. Having an understanding of the community's characteristics can inform broader strategies to reduce readmission rates among patients from "vulnerable communities." Second, we lacked data to fully describe the domains conceptualized as impacting readmission rates. For example, in the domain of social support, several factors known to impact outcomes, including community cohesion and resilience, have not been assessed in this population, which may have underestimated the effect of social support in the model. Third, it is unknown whether these variables would also have the same effect on the readmission rates of other populations, for example, younger or commercially insured patients, or on other outcomes, such as mortality. Finally, we are largely unable to determine the degree to which community factors are contributing above and beyond individual factors because many of these factors (e.g., income, commute to work, health behaviors such as exercise and diet) are not known to us on an individual patient level. As such, it is possible that we are measuring factors that are a proxy for individual risk rather than identifying some intrinsic characteristics of communities that are inherently meaningful. Hopefully, attention to contextual factors will drive more individual and community-level data so that we can best understand patient risk and implement and study strategies to reduce that risk.

## Conclusion

This comprehensive examination of publicly available contextual factors, centered on patient residence, identified community level variables across a number of domains which explained almost a quarter of the variation in hospital-wide readmission rates. These findings provide a number of targets that policymakers, payers, community organizations and hospitals can and should engage with in order to reduce readmission rates and provide safer, more efficient, patient-centered care.

## Acknowledgments

We are grateful for the assistance of Dr. Li Li, PhD, MS–a senior statistician at the Center for Outcomes Research and Evaluation during the time this study was conducted.

## Author Contributions

**Conceptualization:** Erica S. Spatz, Susannah M. Bernheim, Leora I. Horwitz, Jeph Herrin.

**Data curation:** Jeph Herrin.

**Formal analysis:** Erica S. Spatz, Jeph Herrin.

**Investigation:** Erica S. Spatz, Jeph Herrin.

**Resources:** Leora I. Horwitz.

**Supervision:** Leora I. Horwitz.

**Writing – original draft:** Erica S. Spatz.

**Writing – review & editing:** Susannah M. Bernheim, Leora I. Horwitz, Jeph Herrin.

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
