## [Decision Letter · Decision Letter 0]

30 Mar 2020

PONE-D-20-03174

Community Factors and Hospital Wide Readmission Rates: Does Context Matter

PLOS ONE

Dear Dr. Spatz,

Thank you for submitting your manuscript to PLOS ONE. After careful consideration, we feel that it has merit but does not fully meet PLOS ONE’s publication criteria as it currently stands. Therefore, we invite you to submit a revised version of the manuscript that addresses the points raised during the review process.

We would appreciate receiving your revised manuscript by May 14 2020 11:59PM. To enhance the reproducibility of your results, we recommend that if applicable you deposit your laboratory protocols in protocols.io, where a protocol can be assigned its own identifier (DOI) such that it can be cited independently in the future. For instructions see: http://journals.plos.org/plosone/s/submission-guidelines#loc-laboratory-protocols

We look forward to receiving your revised manuscript.

Kind regards,

Prof, Mojtaba Vaismoradi, PhD, MScN, BScN

Academic Editor

PLOS ONE

Journal Requirements:

"All authors work under contract with the Centers for Medicare and Medicaid Services (CMS) to develop and maintain performance measures used in public reporting programs."

Reviewers' comments:

Reviewer #1: Title: Community Factors and Hospital Wide Readmission Rates: Does Context Matter?

This manuscript provides valuable information about community factors relationship with hospital-wide readmission. However, it should be revised professionally. Please find my comments in follow:

General:

The authors should follow the author guideline of PlosOne journal.

Please edit the manuscript in terms of some grammar errors and Punctuation.

Abstract:

You should specify where the study was conducted or which country data was used for analysis.

" R2" should be added for each of percentage of domains.

Introduction:

Page 4, line3: “report”

Paragraph 2 do not have references, please add

Methods:

Article writing should be standard to make it easier for readers to follow the article, so make the following changes

Please move the “conceptual model” to the introduction section

Then in the methods discuss clearly about: 1. Design and Setting

2. Participants

3. Study Variables (including Main variable and Confounders)

Please write the full form of the "RWJF" in first use.

Please develop a section under the subheading of "Ethical Considerations” in the method.

Results:

“R2" should be added for each of percentage of domains

Table 2: More detail about multivariable model should be added including: 1. Describe qualitatively value “1…5”

2. Beta coefficient

3. CI 95%

Discussion:

Please specify and justified the role of race/ethnicity in HWR in relation to minority diminish return theory (MDR) that recently developed by number of scholars such as DR. Assari and Bazargan in US context.

Reviewer #2: The manuscript is generally scientifically sound with information that will contribute to knowledge and with implications for new strategies in patient management to reduce readmissions. The following observations however need to be corrected.

Methods

The study area is left out in the methods. Is the data for the whole of USA?

It is desirable to have examples of such conditions and procedures that were excluded.

Results

Figure 1 not provided in this manuscript

Discussion

Pg 17 - Cite some previous studies that observed social risk factors contribution to readmission rates. Compare findings of this study with previous studies (if there are). It must be clearly stated if there are no previous studies or similar studies to be compared with this study.

---

## [Author Response · Author response to Decision Letter 0]

2 Jun 2020

In addition to an itemized response to the reviewers' comments (submitted in a separate document), we additionally made the following revisions based on the Editor's comments: We updated the formatting style. We also updated our Conflicts of Interest statement, to: “All authors work under contract with the Centers for Medicare and Medicaid Services (CMS) to develop and maintain performance measures used in public reporting programs.” Finally, we will make a minimal dataset available through Open ICPSR (https://www.openicpsr.org/openicpsr/) a repository for sharing study data, should the manuscript be published for publication. We also updated the funding support with the initials of authors funded through this grant, "This study was funded by a grant from the Agency for Healthcare Research and Quality, titled: Understanding Hospital Readmission Rates: Patient, Hospital and Community Effects (1 R01 HS022882 01; Authors SMB, LIH, JH)."

---

## [Decision Letter · Decision Letter 1]

2 Jul 2020

PONE-D-20-03174R1

Community Factors and Hospital Wide Readmission Rates: Does Context Matter

PLOS ONE

Dear Dr. Spatz,

Thank you for submitting your manuscript to PLOS ONE. After careful consideration, we feel that it has merit but does not fully meet PLOS ONE’s publication criteria as it currently stands. Therefore, we invite you to submit a revised version of the manuscript that addresses the points raised during the review process.

We look forward to receiving your revised manuscript.

Kind regards,

Prof, Mojtaba Vaismoradi, PhD, MScN, BScN

Academic Editor

PLOS ONE

Reviewers' comments:

Reviewer #1: Thank you for addressing my comments. I did not find a footnote in table 2 identifying the values 1-5 as representing quintiles of each variable that you mentioned in the reviewer response.

Reviewer #2: No additional new comment for the authors. Authors have addressed comments raised in initial review earlier submitted

---

## [Author Response · Author response to Decision Letter 1]

19 Sep 2020

We updated Table 2 to denote that the values 1-5 represented quintiles. We added a footnote and added a "Q" before the numerical value.

---

## [Editor Report · Decision Letter 2]

23 Sep 2020

Community Factors and Hospital Wide Readmission Rates: Does Context Matter

PONE-D-20-03174R2

Dear Dr. Spatz,

We’re pleased to inform you that your manuscript has been judged scientifically suitable for publication and will be formally accepted for publication once it meets all outstanding technical requirements.

Kind regards,

Prof, Mojtaba Vaismoradi, PhD, MScN, BScN

Academic Editor

PLOS ONE

---

## [Editor Report · Acceptance letter]

13 Oct 2020

PONE-D-20-03174R2 

Community Factors and Hospital Wide Readmission Rates: Does Context Matter? 

Dear Dr. Spatz:

I'm pleased to inform you that your manuscript has been deemed suitable for publication in PLOS ONE. Congratulations! Your manuscript is now with our production department. 

Kind regards, 

on behalf of

Professor Mojtaba Vaismoradi 

Academic Editor

PLOS ONE